# Fracture and Damage Evolution of Multiple-Fractured Rock-like Material Subjected to Compression

**DOI:** 10.3390/ma15124326

**Published:** 2022-06-18

**Authors:** Taoying Liu, Mengyuan Cui, Qing Li, Shan Yang, Zhanfu Yu, Yeshan Sheng, Ping Cao, Keping Zhou

**Affiliations:** 1School of Resources & Safety Engineering, Central South University, Changsha 410083, China; 215511013@csu.edu.cn (M.C.); 215512076@csu.edu.cn (Y.S.); 125501001@csu.edu.cn (P.C.); 217027@csu.edu.cn (K.Z.); 2China Railway Construction Group Corporation Limited, Beijing 100040, China; 18028675688@139.com (Q.L.); 13822887171@139.com (Z.Y.)

**Keywords:** multi-crack interaction, different geometries, crack propagation, failure process, damage and fracture criterion

## Abstract

Multiple compression tests on rock-like samples of pre-existing cracks with different geometries were conducted to investigate the strength properties and crack propagation behavior considering multi-crack interactions. The progressive failure process of the specimens was segmented into four categories and seven coalescence modes were identified due to different crack propagation mechanisms. Ultimately, a mechanical model of the multi-crack rock mass was proposed to investigate the gradual fracture and damage evolution traits of the multi-crack rock on the basis of exploring the law of the compression-shear wing crack initiation and propagation. A comparison between theory and experimental results indicated that the peak strength of the specimens with multiple fractures decreased initially and subsequently increased with the increase in the fissure inclination angles; the peak strength of specimens decreased with the increase in the density of fissure distribution.

## 1. Introduction

Rocks are natural and heterogeneous materials that contain different defect patterns, and the behavior of “rock-like” materials is determined not only by the properties of the intact rock itself, but also by the presence of discontinuities/defects [1,2,3,4,5]. The cracks in the rock mass not only decrease its strength and stiffness, but also bring a source of initiation of new cracks, which, in turn, may propagate and link with other cracks. Understanding the failure process of fractured rock mass subjected to compression is, therefore, essential in rock engineering, mining engineering, deep underground structures, and geotechnical engineering practice [6,7,8,9,10]. In order to study the physical properties, deformation behavior and mechanical mechanism of rock-like materials during the fracturing process, laboratory loading tests have been extensively employed [11,12,13], and comprehensive research has been conducted to develop crack propagation theories [14,15,16,17] or calculate the stress intensity factors at the crack tips [18,19,20,21], as well as quantifying the relationship between micro damage and the macro-mechanical properties of rock mass [22,23,24,25]. Significant advances have been made in understanding the failure process of fractured rock mass [26,27,28].

In practical rock engineering, most rock mass exists with large numbers of cracks that interact with each other and, thus, the crack propagation and coalescence mechanism of multi-crack will be more complex, which is not just material related, but the geometries of the flaws inside the rock also have a key effect on the cracking behavior. For the multi-crack rock mass, a common failure mode is splitting, where the fracture surface is approximately parallel to the direction of the applied loading. At the early stages, when the wing crack is short or the crack space is relatively small, the growth is usually dominated by the stress field around the pre-existing fracture. As the crack extends, the interaction between cracks leads to a damaged connection and unstable break in the rock bridge. The interaction theory of multiple cracks in rock masses is a key factor in the analysis of their “micro-damage”, and most of the previous studies on rocks containing multiple cracks were based on the dilute distribution condition that the crack interaction is neglected, with very limited works so far being conducted exploring the influences of crack inclination and crack number. Besides, most theoretical research on the failure mechanism of fractured rocks is mainly focused on an individual crack, and few models have been reported for revealing the evolution laws of the stress intensity factor at the tip of a wing crack in multiple-fractured rock mass.

Therefore, for a better understanding of the fracture and damage mechanisms in multiple-crack rock-like materials, multiple compression tests on specimens with different geometries of pre-existing cracks were conducted to investigate the crack coalescence laws and strength properties of rock-like materials, considering multi-crack interactions. Meanwhile, the application of the fracture mechanics theory to rock material with cracks is presented. This paper, for the first time, proposes a damage fracture mechanical model of multiple-fractured rock mass, and a new crack initiation criterion is established.

## 2. Experimental Scheme

This rock-like material was made of a mixture of water, white cement, and silica sand, and the dimensions of the specimens were 200 mm × 150 mm × 30 mm. Pre-existing cracks were made by inserting iron sheets (0.4 mm × 20 mm) at the early stage of sample preparation, which were removed as the mortar hardened. Details of each specimen are listed in Figure 1 and Table 1, where *α* is the crack angle, and *β* is the angle of the rock bridge. The physical parameters of the intact specimens are shown in Table 2. The tests were performed on a servo-controlled uniaxial loading apparatus, as shown in Figure 2. The loading rate was 200 N/s. During the loading process, failure patterns and stress–strain curves were recorded by camera and strain gauges, and a clock gauge was used to measure its transverse deformation.

## 3. Multiple-Cracks Failure Mode

### 3.1. Crack Coalescence Mode

A typical crack pattern in compression is illustrated in Figure 3. There are both wing cracks and secondary cracks produced at the tips of pre-existing cracks, subjected to compression. Wing crack is a tensile crack initiating at an angle from the tips of the crack and propagating parallel to the direction of the applied loading. Secondary crack is a shear crack appearing after the formation of the wing crack and it propagates nearly coplanar to the main crack, which is often responsible for specimen failure. When the prefabricated crack angle is at a low value, wing cracks will expand or coalesce approximately in the direction parallel to the maximum principal stress until the failure of the specimen. There is no friction trace on the transfixion plane by observing the failure pattern, as shown in Figure 4. With the increase in the angle of the rock bridge, the rock mass undergoes second crack shear failure, and there is an obvious friction trace on the transfixion plane, as shown in Figure 5. With reference to different crack coalescence mechanisms, several coalescence modes can been identified, as shown in Table 3.

### 3.2. Multiple-Crack Propagation

The failure process of the multiple-fractured rock-like specimens with different geometries is shown in Figure 6, Figure 7, Figure 8 and Figure 9. These figures demonstrate that the crack angle plays a leading role in influencing the micro-crack fracturing and failure mode. The influence of crack density, however, depends on the crack angle.

For the specimens with five cracks, shown in Figure 6, the wing crack expanded gradually till the destruction of the whole specimen as the load increased. Compared with other experiment groups, the fracture mode of the specimens with crack angles of 25° and 45° were almost the same. There was a wing crack produced at the tip of the inclined crack, then the secondary crack appeared on both sides when there was a barrier for wing crack expansion. As the load increased, the wing crack expanded gradually till the specimen horizontal transfixion failure occurred. The specimens with crack angles equal to 75° appeared to have similar failure progress, but as the rock bridge angle increased in magnitude, a second inclined crack appear later, and the horizontal transfixion failure occurred faster. In addition, it would be hard to find the micro-crack extension and penetration phenomena at the crack tip when the angle crack is 90° under this given crack distribution.

As the crack number increased to 10, as shown in Figure 7, the failure occurred based on the secondary inclined crack’s initiation and propagation and the crack was linked to other pre-existing cracks in the same row. At the initial stages of loading, the rock bridge was long, and the wing crack propagated as a single fracture. As the load increased, a coalescence crack occurred. Compared with other experiment groups, the failure modes with crack angles of 25° and 45° were almost the same. For the specimens with crack angles of 75° and 90°, as the crack angle was much larger, the second inclined crack expanded in the vertical direction, and it would be hard to identify the micro-crack extension and penetration phenomena at the crack tip in this pattern.

When the number of cracks increased to 15, as shown in Figure 8, different failure patterns were observed. During the initial loading stages, the wing crack appeared at the tip of the crack, and subsequently, a secondary inclined crack’s initiation occurred, which propagated as the load increased. In this case, the cracks at different rows connected till the failure of the specimen. Compared with other experiment groups, the fracture mode of the specimens with crack angles of 25° and 45° were almost the same. For the specimens with crack angles equal to 75°, the wing crack mainly emerged at the tip of the crack in two sides of the specimen. In this case, there was no wing crack appearing in the center of the specimen, and it would be hard to find the micro-crack extension and penetration phenomena at the tip of the crack when the crack angle was 90°.

With an increasing number of cracks, as shown in Figure 9, the length of the rock bridge between different rows decreased. The wing crack emerged earlier compared with the specimen with 15 cracks, and there were lesser secondary inclined cracks produced. As the load increased, the wing crack on the diagonal between different rows with the same crack direction expanded faster and coalesced. Compared with other experiment groups, the fracture mode of the specimens with crack angles of 25° and 45° were very much the same, and for the specimens with crack angles of 75°, the wing crack mainly emerged at the tip of the crack in two sides of the specimen and caused the specimen’s failure. However, there were some wing cracks that appeared in the center of the specimen. Similar to the previous cases, it would be hard to find the micro-crack extension and penetration phenomena at the tip of the crack when the crack angle is 90°.

### 3.3. Multiple-Crack Failure Mode

There are seven coalescence modes between two pre-existing cracks, owing to the different crack angle and crack number, the multi-crack specimens presented with different failure modes.

(1)Wing crack tension failure mode. The main feature of this mode is that the wing cracks initiate and propagate from the pre-existing crack tips at any particular row towards the crack tips in the same rows. Even though both tensile and shear wing cracks existed, the majority belonged to tensile failures.(2)Second crack shear failure mode. The main feature is that the second shear cracks initiate and propagate from the pre-existing crack tips towards the adjacent row. Again, even though there are both tensile and shear cracks, the majority belong to shear failure.(3)Stepped path failure mode. The main feature is that there are second cracks initiating from the pre-existing crack tips, and subsequently, tension cracks initiate and propagate from the second cracks towards the adjacent row; the combination of these crack results in several “stepped coalescence patterns”, formed on the failure plane.(4)Intact failure mode. The main feature is that the cracks initiate and propagate in the intact material, between or along the cracks. This failure mode usually occurs when the crack angle is 90° and is similar to the failure pattern of the intact specimens.

## 4. Deformation and Strength

### 4.1. Stress–Strain Curves

The stress–strain curves during the failure process are shown in Figure 10, Figure 11, Figure 12 and Figure 13; there are micro-cracks initiating and propagating as the load increases, and the stress–strain curve showed a post-peak softening, and this softening behavior had a direct correlation with the crack development. The crack density merely influenced the transfixion pattern. Compared to the wing crack failure, a rubbing effect was observed on the shearing surface of the specimens, and there was no major strength degradation, especially for specimens with a crack angle of 45°. Meanwhile, the increase in the crack angle resulted in less intense post-peak softening behavior, the stress–strain curves changed from multi-peak values to single-peak values and the specimens led to brittle failure, but the ductility of the specimen was reinforced.

### 4.2. Strength Characteristics

The developed cracks will lead to strength weakening. The relation curves of the peak stress with different numbers of fissures and angles of the cracks are obtained, as shown in Figure 14 and Figure 15. These data suggest that the peak strength of the specimens with multiple fissures decreased initially and then increased with increasing fissure inclination angle. When the fissure inclination angle was 45°, the peak strength was the minimum, and the strength reduction in these specimens was of the order of 60% compared with the intact ones; when the inclination angle increased, the strength increased with the increase in the crack angle. Meanwhile, the increase in the density of fissure distribution would result in a decrease in the peak strength because it is a process of evolution and accumulation of damage to the rock mass as the crack number increased progressively.

## 5. Multi-Crack Damage Fracture Modes

### 5.1. Crack Initiation

It has been shown that wing cracks expand approximately in a direction parallel to the maximum principal stress under the action of external forces [29,30], as shown in Figure 16.

According to the theory of fracture mechanics of materials, the normal stress *σ_ne_* and the shear driving force *τ_en_* on the crack plane are derived [31]:(1)$$\sigma_{ne}=\sigma \sin^{2}\psi$$
(2)$$\tau_{en}=\sigma \sin\psi \cos\psi$$

Friction $\mu \sigma_{ne}+C$ is generated because of the partly closed cracks; the effective shear stress and normal stress are shown, respectively, as:(3)$$\tau_{eff}=\left(1-C_{v}\right)\frac{\sigma }{2}\sin2\psi -\mu \sigma_{ne}-C$$
(4)$$\sigma_{ne}=\sigma_{n}=\left(1-C_{n}\right)\sigma \sin^{2}\psi$$
(5)$$\begin{array}{c} C_{n}=\frac{\pi a}{\pi a+\frac{E_{0}}{\left(1-v_{0}^{2}\right)K_{n}}}\\ C_{v}=\frac{\pi a}{\pi a+\frac{E_{0}}{\left(1-v_{0}^{2}\right)K_{s}}} \end{array}\}$$

Then, the stress intensity factor at the tip of the crack is obtained according to the rock fracture mechanics theory [32]:(6)$$K_{Ι}=\frac{3}{2}\tau_{eff}\sqrt{\pi a}\sin\theta \cos\frac{\theta }{2}$$

The cracking angle *θ* = 75° was obtained by supposing the wing crack propagates along the maximum value. The stress intensity factor can be formulated as:(7)$$K_{Ι}=\frac{2}{\sqrt{3}}\tau_{eff}\sqrt{\pi a}$$

### 5.2. Wing Crack Propagation

Based on the fracture mechanics theory, the stress intensity factor at the wing crack tip can be simplified into the superposition of the two stress intensity factors, as shown in Figure 17:(8)$$K_{Ι}=K_{Ι}^{\left(1\right)}+K_{Ι}^{\left(2\right)}$$

The stress intensity factor t $K_{Ι}^{\left(1\right)}$ is generated by the wing crack, and $K_{Ι}^{\left(2\right)}$ is introduced by the effective shear stress on the main crack with a length of 2*al_ty_*. Based on the rock fracture mechanics theory, the values of the two stress intensity factors are, respectively,
(9)$$K_{Ι}^{\left(1\right)}=\frac{1}{2}\left[\sigma +\sigma \cos2\left(\theta +\beta \right)\right]\sqrt{\pi l}$$
(10)$$K_{ΙΙ}=2\tau_{eff}\sqrt{\frac{al_{ty}}{\pi }}\sin^{-1}\left(\frac{1}{l_{ty}}\right)$$

*K*_I_(*θ*) produced by stress *σ_θ_* in the direction of *θ* is [33]:(11)$$K_{Ι}\left(\theta \right)=\frac{3}{2}K_{ΙΙ}\sin\theta \cos\frac{\theta }{2}$$

$K_{Ι}^{\left(2\right)}$ is calculated as:(12)$$K_{Ι}^{\left(2\right)}=3\tau_{eff}\sqrt{\frac{al_{ty}}{\pi }}\sin^{-1}\left(\frac{1}{l_{ty}}\right)\sin\theta \cos\frac{\theta }{2}$$

Thus, *K*_I_ can be expressed as:(13)$$K_{Ι}=3\tau_{eff}\sqrt{\frac{al_{ty}}{\pi }}\sin^{-1}\left(\frac{1}{l_{ty}}\right)\sin\theta \cos\frac{\theta }{2}-\frac{1}{2}\left[\sigma +\sigma \cos2\left(\theta +\beta \right)\right]\sqrt{\pi l}$$

Therefore, the revised Horii and Nemat–Nasser wing crack model can be seen below [34]:(14)$$K_{Ι}=\frac{2a\tau_{eff}\sin\theta }{\sqrt{\pi \left(l+0.27a\right)}}-\frac{1}{2}\left[\sigma +\sigma \cos2\left(\theta +\beta \right)\right]\sqrt{\pi l}$$

Comparing Equation (13) with Equation (14), the following equations were obtained: (15)$$3\tau_{eff}\sqrt{\frac{al_{ty}}{\pi }}\sin^{-1}\left(\frac{a}{al_{ty}}\right)\sin\theta \cos\frac{\theta }{2}=\frac{2a\tau_{eff}\sin\theta }{\sqrt{\pi \left(l+0.27a\right)}}$$

As l l→0, the following equation was obtained:(16)$$l_{ty}=\frac{0.667}{\cos^{2}\left(0.5\theta \right)}$$

As l→∞, the following equation was obtained:(17)$$l_{ty}=1+\frac{9l\cos^{2}\frac{\theta }{2}}{4a}$$

Comparing Equation (15) with Equation (16), *l_ty_* can be set as
(18)$$l_{ty}=\left[1+\frac{9l}{4a}\cos^{2}\left(\frac{\theta }{2}\right)\right]\left(1-e^{-\frac{1}{a}}\right)+0.667\sec^{2}\left(\frac{\theta }{2}\right)e^{-\frac{1}{a}}$$

### 5.3. The Multi-Crack Interaction Models

For multiple-fractured rock-like material, the interaction between the cracks will lead to damage on the rock bridge [35], as shown in Figure 18.

Assuming the number of the cracks per unit area is *N_A_* in Figure 18, the distance between the cracks *S* and the rock bridge length between the wing cracks *T* is given as follows:(19)$$\begin{array}{c} S=\frac{1}{\sqrt{N_{A}}}\\ T=S-2\left(l+a\cos\psi \right) \end{array}\}\;$$

$\sigma_{3}^{\prime }$ was applied on the rock bridge:(20)$$\sigma_{3}^{\prime }=\frac{T_{e}\sin\psi }{S-2\left(l+a\cos\psi \right)}$$
where $T_{e}=2a\tau_{eff}$.

$\sigma_{3}^{\prime }$ will produce an additional intensity factor:(21)$$K_{Ι}^{\prime }=-\sigma_{3}^{\prime }\sqrt{\pi l}=\frac{a\sigma sin2\psi \sin\psi }{N_{A}^{-1/2}-2\left(l+a\cos\psi \right)}\sqrt{\pi l}$$

Considering the multi-crack interaction, the stress intensity factor as wing crack propagation is established by combining Equations (10) and (11):(22)$$K_{Ι}=K_{Ι}+K_{Ι}^{\prime }=3\tau_{eff}\sqrt{\frac{al_{ty}}{\pi }}\sin^{-1}\left(\frac{1}{l_{ty}}\right)\sin\theta \cos\frac{\theta }{2}-\frac{1}{2}\left[\sigma +\sigma \cos2\left(\theta +\beta \right)\right]\sqrt{\pi l}+\frac{a\sigma sin2\psi \sin\psi }{N_{A}^{-1/2}-2\left(l+a\cos\psi \right)}\sqrt{\pi l}$$

When *K*_I_ = *K*_IC_ in Equation (12), the cracks begin to expand. Figure 19 shows the interaction of the multi cracks makes the stress intensity factor at the crack tip larger than that related to a single wing crack.

Figure 20 presents the initial crack strength curves with different fissure crack angles. It can be seen that with the increasing inclination angle of the fissures, the initial crack strength decreased initially and then increased.

### 5.4. Multi-Crack Damage Models

The crack quantity is used to define *D*_0_ for initial damage and *D* for damage when the wing crack length expends to *l* [36]:(23)$$\begin{array}{c} D_{0}=\pi {\left(a\cos\psi \right)}^{2}N_{A}\\ D=\pi {\left(l+a\cos\psi \right)}^{2}N_{A} \end{array}\}$$

Substituting Equation (13) into Equation (10), the following equation is obtained,
(24)$$\sigma_{3}^{\prime }=-\frac{\tau_{en}\tan\psi {\left(D_{0}/\pi \right)}^{1/2}}{a-2a{\left(D/\pi \right)}^{1/2}}$$

Combining Equations (10), (13) and (14), the relation curves between the damage variables *D* and stress intensity factor $\left(K_{Ι}/\sigma_{Ι}\sqrt{\pi a}\right)$ with different crack densities are shown in Figure 21.

Figure 21 shows that with the increase in the damage variable, the stress intensity factor $K_{Ι}/\sigma_{Ι}\sqrt{\pi a}$ decreases gradually. The more scattered the cracks are, the higher the stress intensity of the wing crack will be; as the damage variable *D* changed from *D*_0_ to 1, the multiple-fractured rock-like material lost its strength.

## 6. Conclusions

(1)Compression experiments were conducted for brittle rock-like samples with multiple fissures to explore the failure rules of the rock-like material with different fissure inclination angles and density distributions. Owing to the differences in crack geometries, seven coalescence modes were identified and the failure process of the multiple-fractured specimens was divided into four categories. It was also found that the strength of the multiple-fractured specimens was affected by the crack angle and crack number, and the crack angle was the main influencing factor. The crack density merely affected the transfixion pattern, and the peak value decreased with the increase in the crack number.(2)Based on the rock fracture mechanics theory, a wing crack propagation model considering the interaction of multiple cracks was established, for the first time, in this study. The multi-crack effect would result in “reinforcement” of the wing crack’s propagation. The multiple-crack initiation criterion was further developed to predict the propagation process of the fractures in the rock mass, which provided a theoretical basis for applications in rock engineering. Comparisons between theory and experiment results indicated that the peak strength of specimens with multiple cracks decreased initially and increased with the increasing inclination angle of the fissures, but the peak strength decreased with the increasing fissure distribution density. This work provides a basis for quantitative research on fractured rock-mass failure subjected to compression.

## Figures and Tables

**Figure 1 materials-15-04326-f001:**
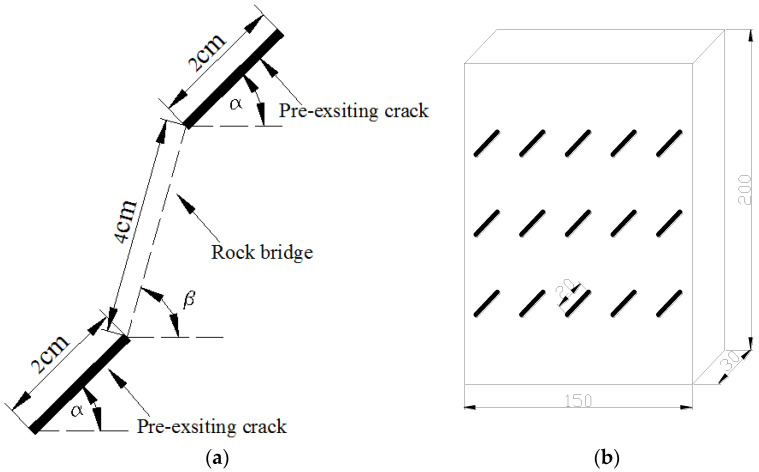
Multiple-cracked rock-like material used in the present study. (**a**) Two cracks; (**b**) Multiple cracks.

**Figure 2 materials-15-04326-f002:**
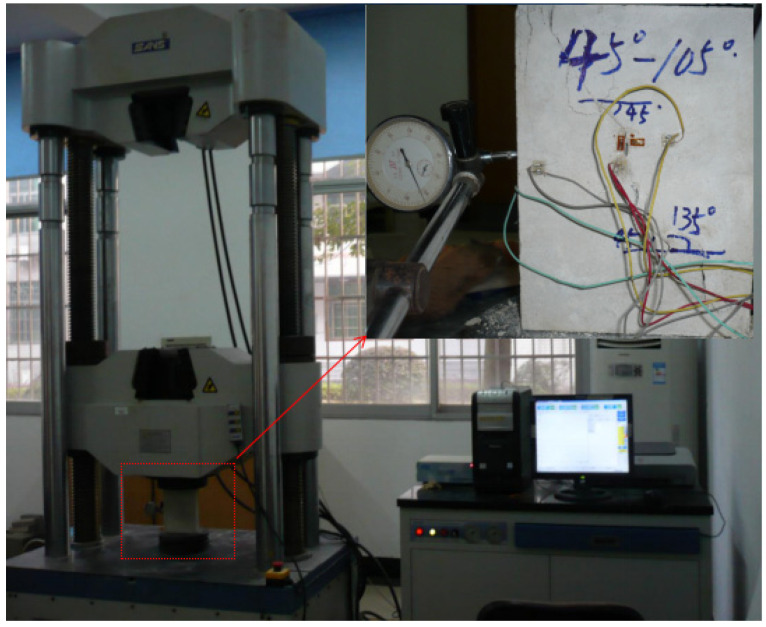
Experimental setup for multiple-fractured rock-like material.

**Figure 3 materials-15-04326-f003:**
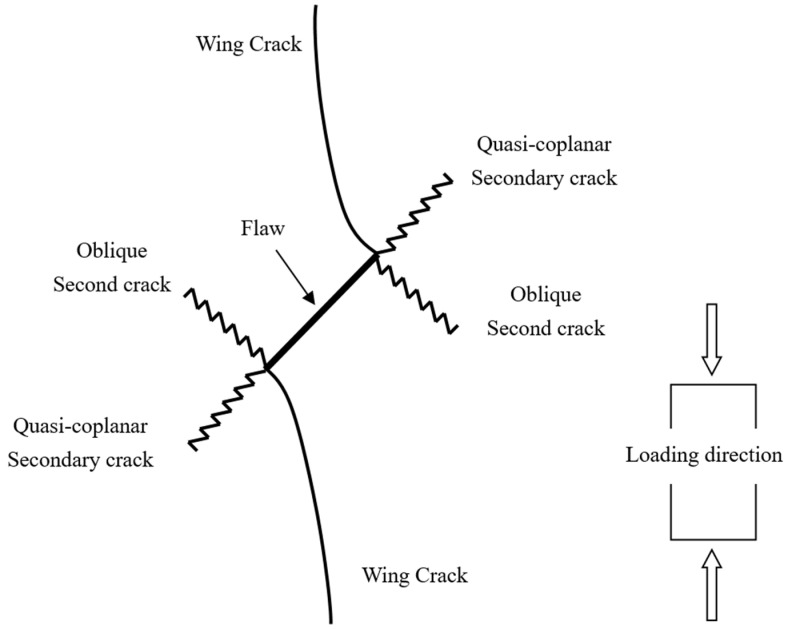
Crack propagation from pre-existing crack subjected to compression.

**Figure 4 materials-15-04326-f004:**
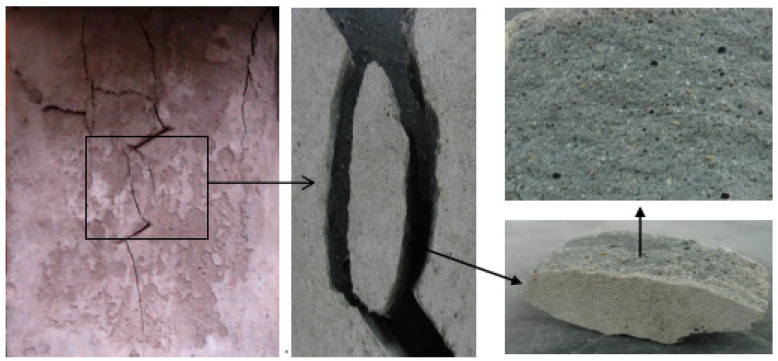
Wing crack propagation failure and its pattern.

**Figure 5 materials-15-04326-f005:**
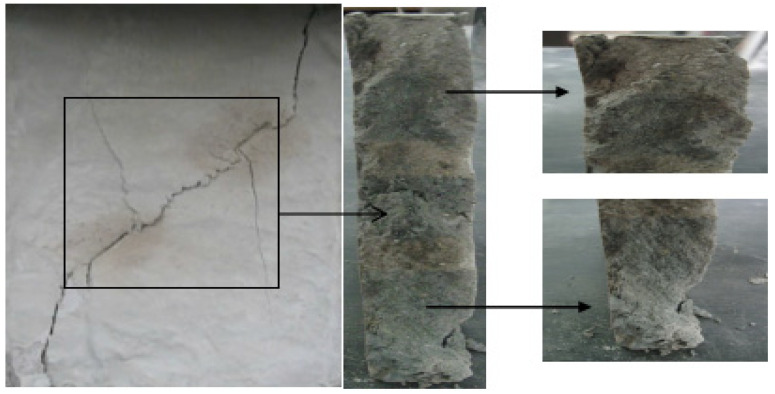
Second crack shear failure and its pattern.

**Figure 6 materials-15-04326-f006:**
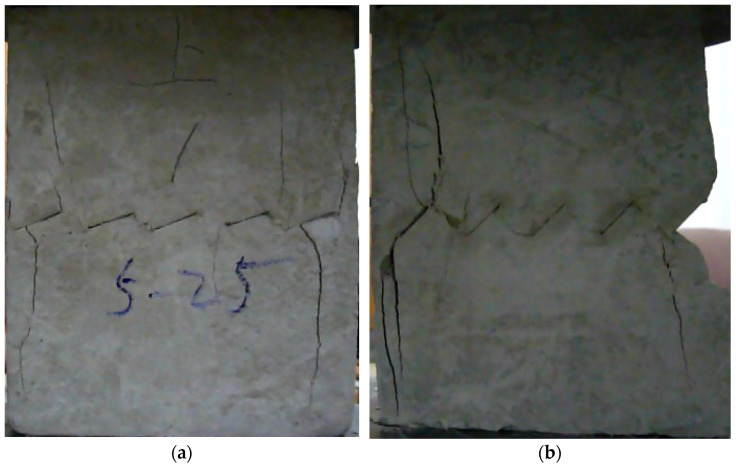
The failure process with 5 cracks. (**a**) 25°; (**b**) 45°; (**c**) 75°; (**d**) 90°.

**Figure 7 materials-15-04326-f007:**
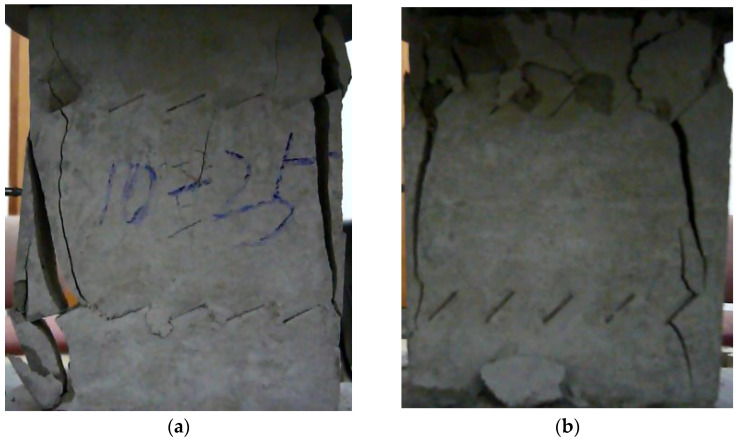
The failure process with 10 cracks. (**a**) 25°; (**b**) 45°; (**c**) 75°; (**d**) 90°.

**Figure 8 materials-15-04326-f008:**
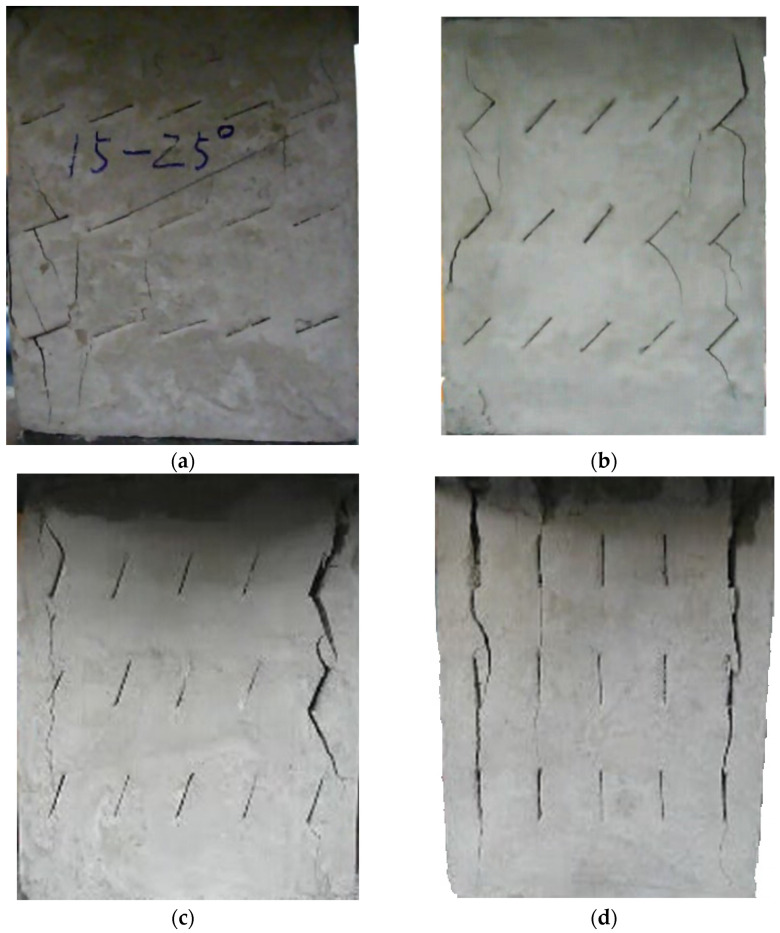
The failure process with 15 cracks. (**a**) 25°; (**b**) 45°; (**c**) 75°; (**d**) 90°.

**Figure 9 materials-15-04326-f009:**
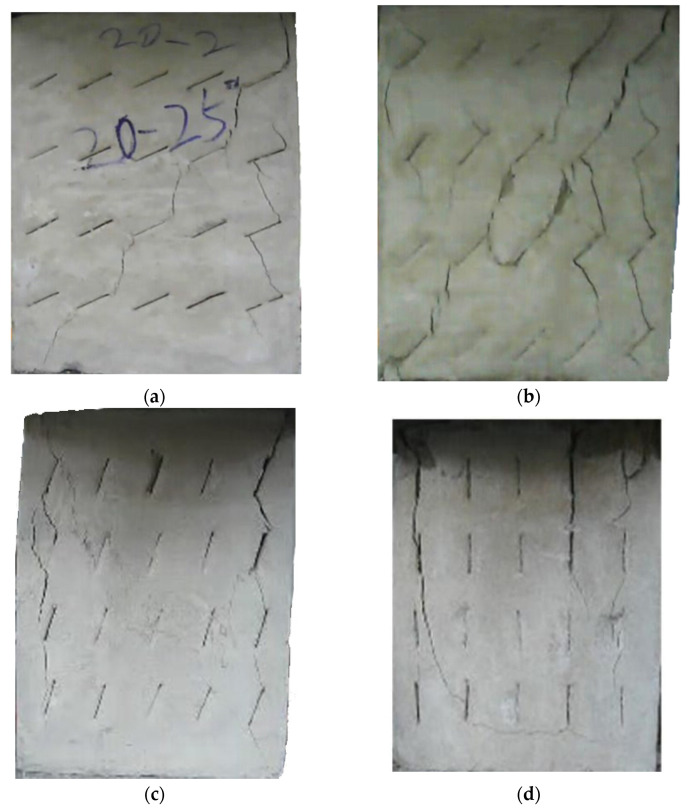
The failure process with 20 cracks. (**a**) 25°; (**b**) 45°; (**c**) 75°; (**d**) 90°.

**Figure 10 materials-15-04326-f010:**
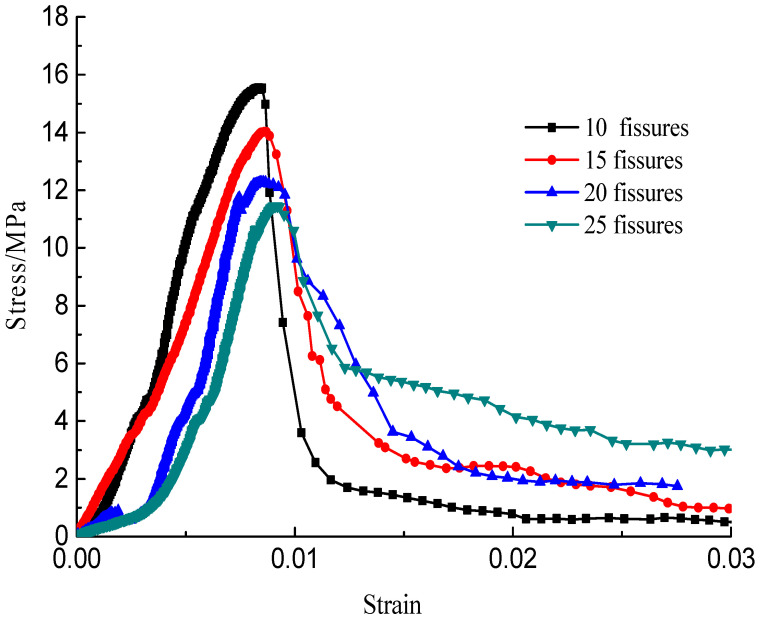
Stress–strain curves with a crack angle of 25°.

**Figure 11 materials-15-04326-f011:**
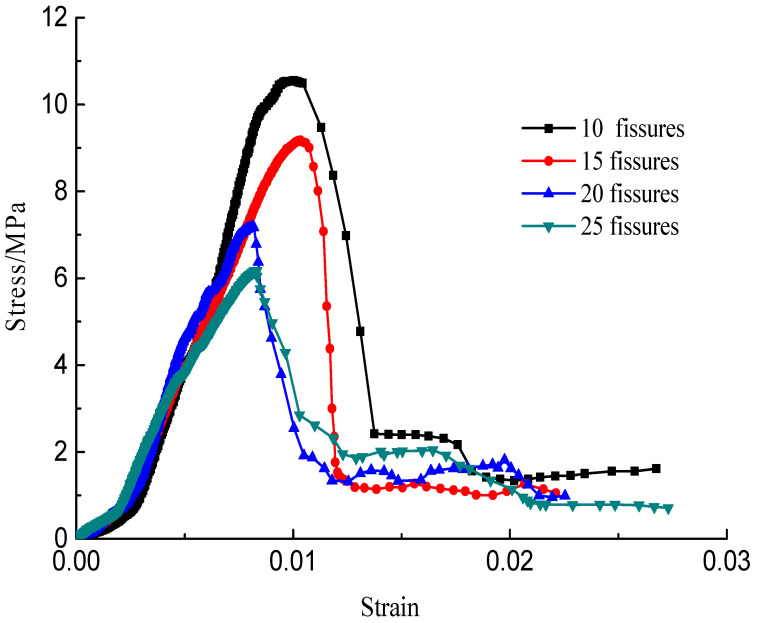
Stress–strain curves with a crack angle of 45°.

**Figure 12 materials-15-04326-f012:**
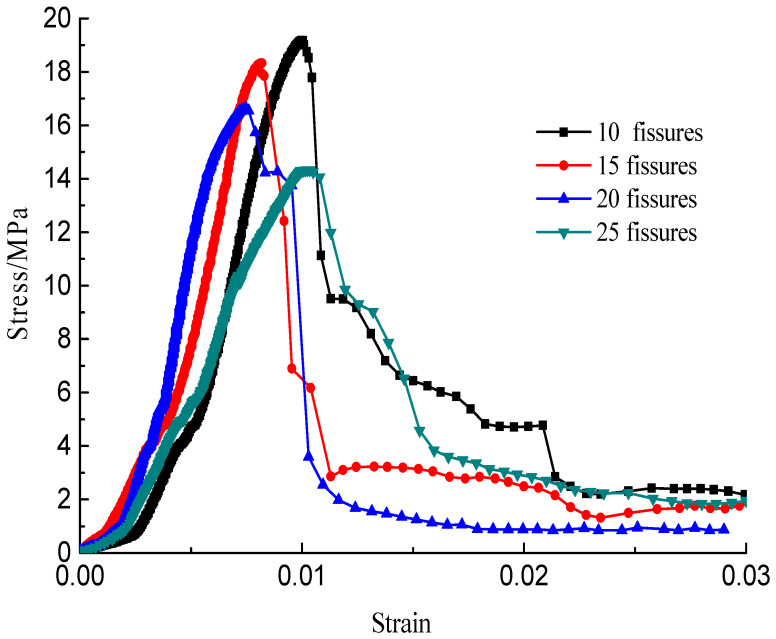
Stress–strain curves with a crack angle of 75°.

**Figure 13 materials-15-04326-f013:**
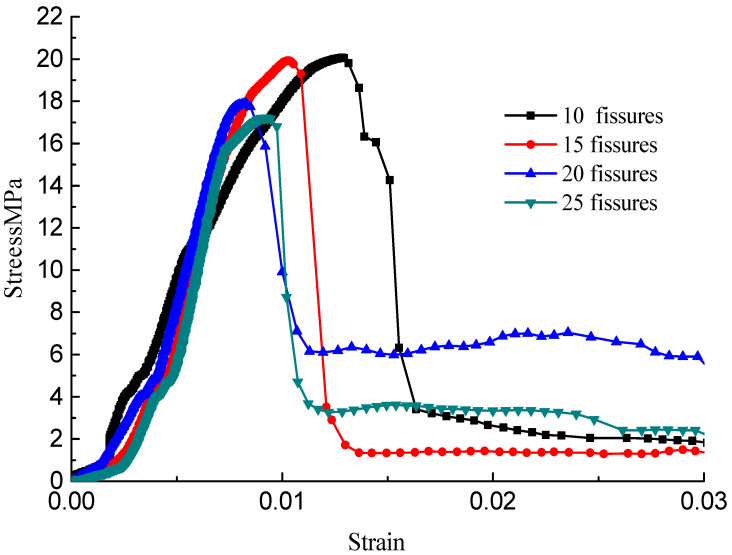
Stress–strain curves with a crack angle of 90°.

**Figure 14 materials-15-04326-f014:**
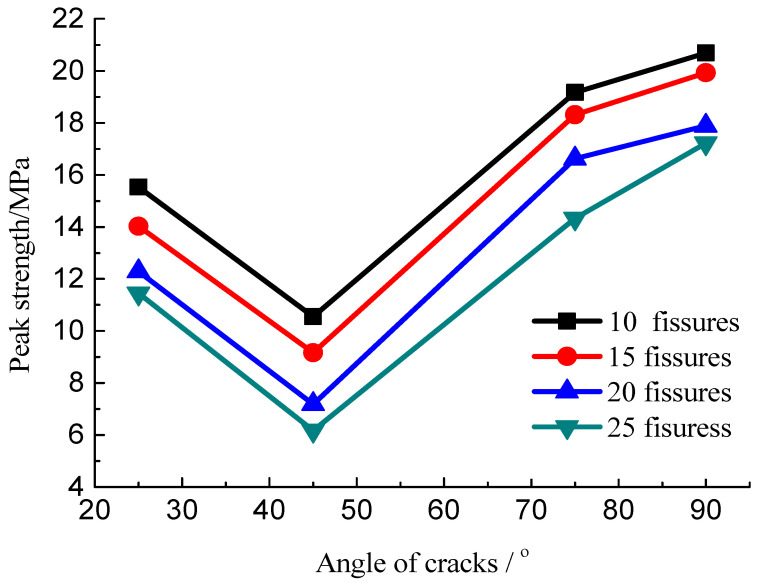
Relation curve of the peak stress with different number of cracks.

**Figure 15 materials-15-04326-f015:**
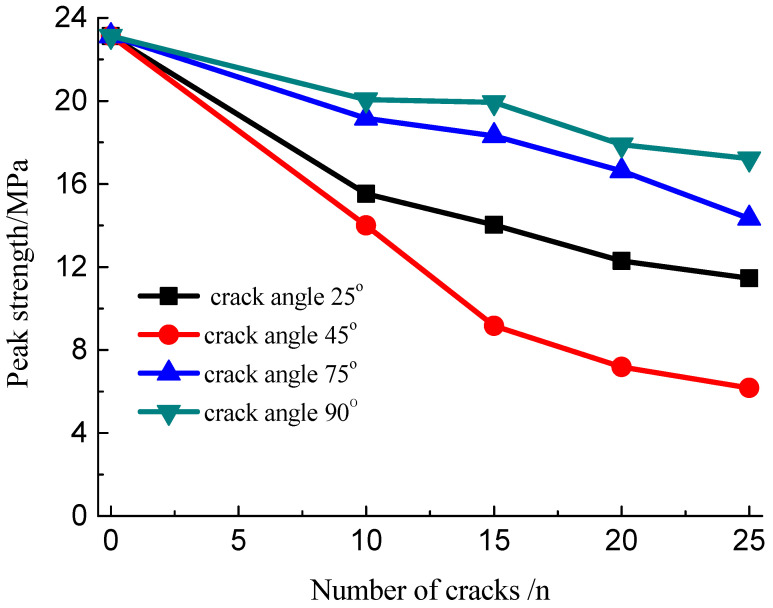
Relation curve of the peak stress under different crack angles.

**Figure 16 materials-15-04326-f016:**
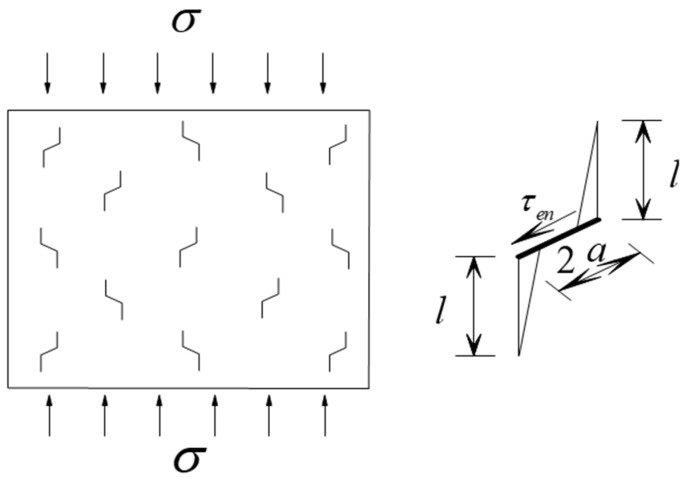
Wing crack propagation subjected to compression.

**Figure 17 materials-15-04326-f017:**
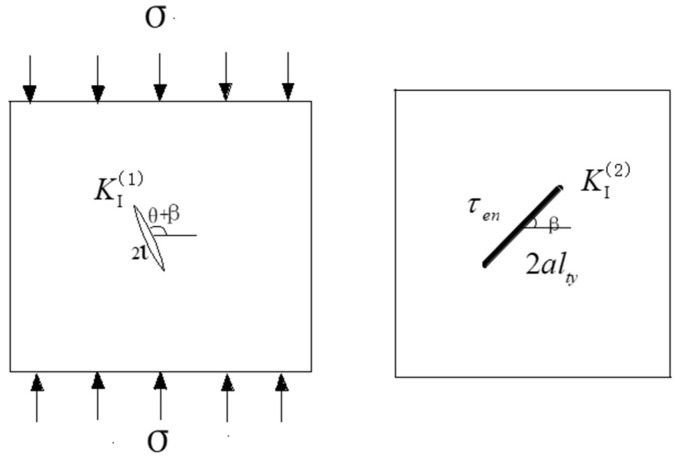
Stress intensity factor as wing crack propagation.

**Figure 18 materials-15-04326-f018:**
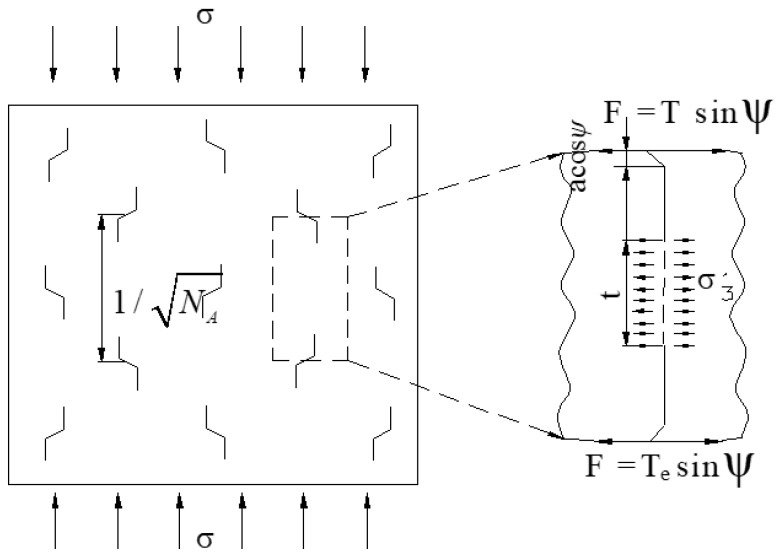
Multiple interacting cracks.

**Figure 19 materials-15-04326-f019:**
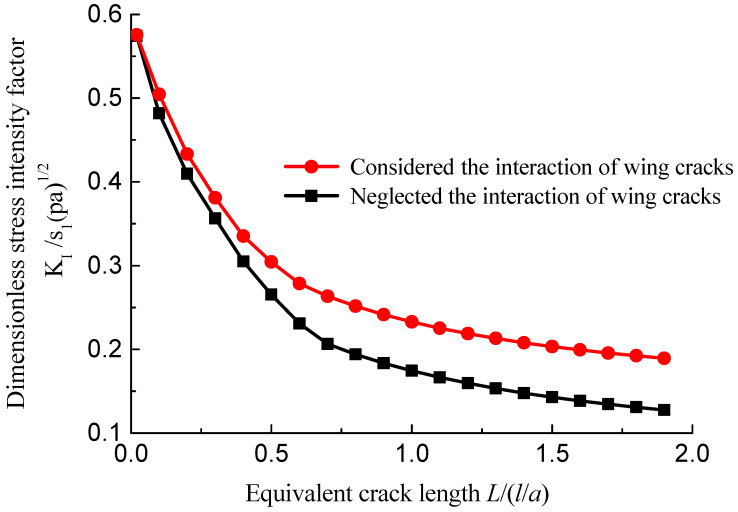
Comparison of stress intensity factor at the wing crack tip considering the interaction of multiple wing cracks.

**Figure 20 materials-15-04326-f020:**
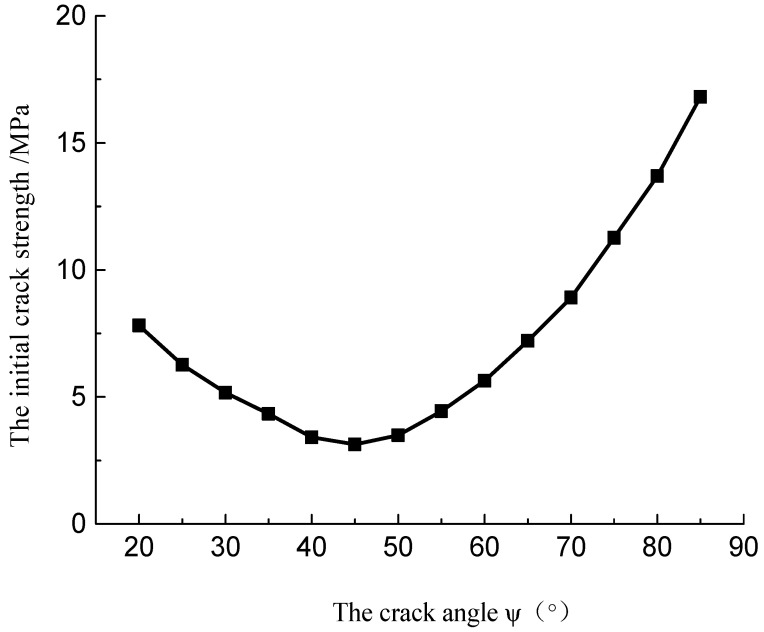
The relationship between crack initiation strength and crack angle.

**Figure 21 materials-15-04326-f021:**
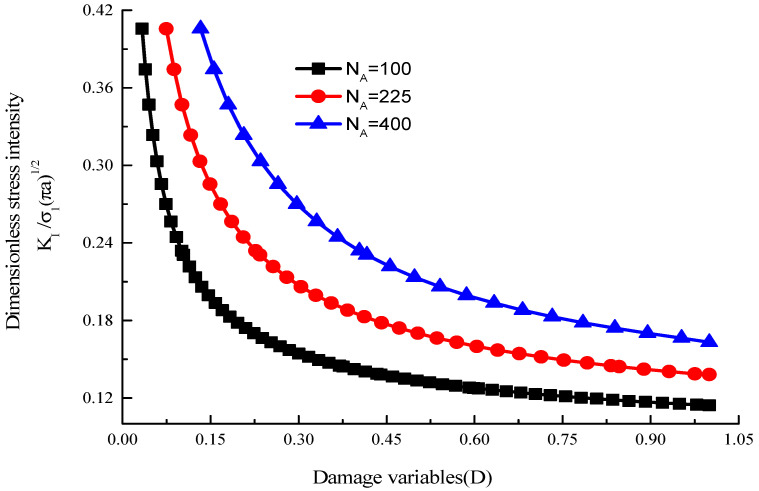
The relationship between damage variables and stress intensity factor.

**Table 1 materials-15-04326-t001:** Details of specimens containing multiple cracks.

Number of Cracks	Crack Angle/°	Angle of Rock Bridge *β*/°
Two cracks	*α* = 25	25, 45, 75, 90, 105
	*α* = 45	45, 75, 90, 105
	*α* = 75	75, 90, 105
Multiple cracks	*α* = 25	5, 10, 15, 20, 25
	*α* = 45	5, 10, 15, 20, 25
	*α* = 75	5, 10, 15, 20, 25
	*α* = 90	5, 10, 15, 20, 25

**Table 2 materials-15-04326-t002:** Mechanical parameters of specimens.

UCS/MPa	UTS/MPa	Density g/cm^3^	Young’s Modulus/MPa	Poisson’s Ratio
23.1	2.8	2.019	2.3 × 10³	0.23

**Table 3 materials-15-04326-t003:** Crack coalescence modes subjected to compression from the experimental observations.

Failure Modes	Failure Process	Failure Characteristics
Wing crack propagation failure	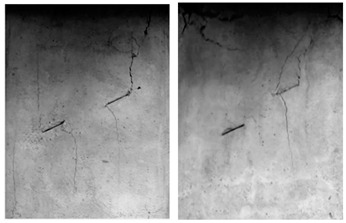	There are wing cracks which initiate at the tip of two cracks or only one crack, and propagate in the direction parallel to the maximum principal stress.
Wing crack coalescence failure	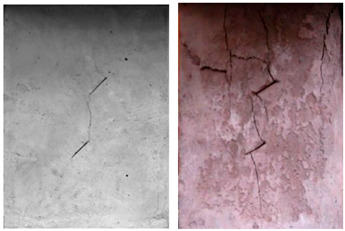	The wing cracks connect to another wing crack or connect to pre-existing crack when it extends to a certain length.
Second crack shear failure	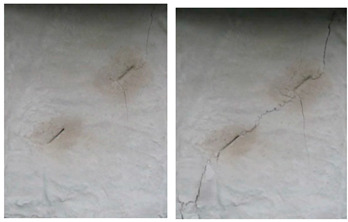	There are second cracks produced from the pre-existing cracks which finally connect; this usually occurs when the cracks are co-planar.
Tension shear combined fracture failure	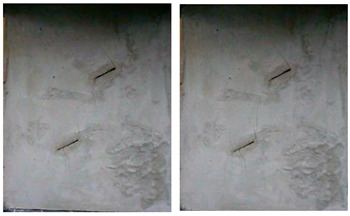	When the wing cracks expand to a certain length, the rock bridge between adjacent wing cracks is cut off in the shear direction.
Wing crack and second crack connection failure	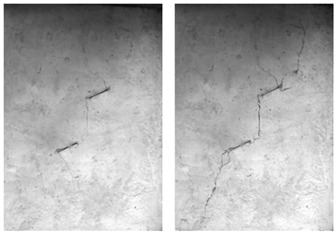	The wing crack developed at one crack tip is connected to the second crack.
Wing crack shear connection failure	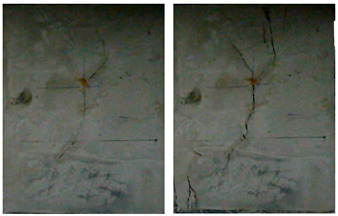	The wing crack expands under loading and connects to another pre-existing crack.
Brittle failure	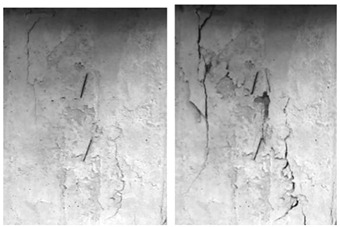	Although there are many wing cracks or second cracks appearing during the loading process, they are not related to the specimen’s failure.

## Data Availability

Data sharing is not applicable in this article.

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
