# Peer review of "Fracture and Damage Evolution of Multiple-Fractured Rock-like Material Subjected to Compression"

_materials, 2022, doi:10.3390/ma15124326_

Round 1

Reviewer 1 Report

The paper reports the "Fracture and damage evolution of multiple rock-like material cracks in compression".

 the investigation is interesting, but the paper should be improved. please follow the below feedback to improve the paper.

1- The abstract section should be improved. Important results should be highlighted.

2- The introduction section should be improved. In the discussion of previous studies, the gap of them that resulted in doing the current study should be highlighted. and please remove the old references from 1980-1998, and use new 10 references from 2018-2021.

3- The novelty of this study is not so clear. So, the novelty of this study should be highlighted in more details by discussing the weakness of previous investigations. This is must improved it.

4- The obtained results should be further elaborated and interpreted to provide more significant and important findings and conclusions. This apply to all results and figures discussion. 

5- Moderate English changes required.

Reviewer 2 Report

The paper ”Fracture and damage evolution of multiple rock-like material cracks in compression” is suitable for publication after some minor corrections. The authors should improve table 1 with a new arrangement because it does not fit well on pages 3-4. Also, it would be suitable that the authors to present some optical (more the 100X magnitude) or SEM images with the cracks at the breaking area. The author's contribution is missing. In rest is ok.

Author Response

请参阅附件。

Reviewer 3 Report

The authors have carried out a fracture and damage analyses of the evolution of multiple cracks when rock-like specimens are submitted to compression. The work presents an experimental study and the subject is in the scope of the journal. I would like to suggest the authors to address the following queries for further clarifications: 

  1. The objectives of the study must the clarified in the Introduction section. I believe Fig 1 can be moved into the Materials section for a better explanation of theories and methodologies.
  2. Angles in Table 1 may be written in a column format rather then into rows.
  3. "3. Multiple cracks failure mode" Section: figures of the cracks pattern can be evaluated in a quantitative approach by some image processing, regarding extension and orientation.
  4. It is said p.4 line 23-24 "...During the loading process, failure patterns and stress-strain curves of specimens were recorded.." more details need to be presented how strain for instance has been measured? Do the authors have used any image processing/correlation technique for instance?
  5. After Eqs. (4) or (5) for instance, the paragraph must be removed.

Author Response

请参阅附件。

Round 2

Reviewer 1 Report

Accepted in current version 

Author Response

Thank you very much for your positive comments. We greatly appreciate your time and efforts to improve our manuscript.

Reviewer 3 Report

I believe the authors have improved the revised manuscript accordingly.

Author Response

(The authors gave the same response as above.)
